# Alkali metal reduction of alkali metal cations

Kyle G. Pearce [1], Han-Ying Liu [1], Samuel E. Neale [1], Hattie M. Goff[1], Mary F. Mahon[1], Claire L. McMullin [1] ✉ & Michael S. Hill [1] ✉

Counter to synthetic convention and expectation provided by the relevant standard reduction potentials, the chloroberyllate, [{SiN$^{Dipp}$}BeClLi]$_2$ [{SiN$^{Dipp}$} = {CH$_2$SiMe$_2$N(Dipp)}$_2$; Dipp = 2,6-$i$-Pr$_2$C$_6$H$_3$)], reacts with the group 1 elements (M = Na, K, Rb, Cs) to provide the respective heavier alkali metal analogues, [{SiN$^{Dipp}$}BeClM]$_2$, through selective reduction of the Li$^+$ cation. Whereas only [{SiN$^{Dipp}$}BeClRb]$_2$ is amenable to reduction by potassium to its nearest lighter congener, these species may also be sequentially interconverted by treatment of [{SiN$^{Dipp}$}BeClM]$_2$ by the successively heavier group 1 metal. A theoretical analysis combining density functional theory (DFT) with elemental thermo-chemistry is used to rationalise these observations, where consideration of the relevant enthalpies of atomisation of each alkali metal in its bulk metallic form proved crucial in accounting for experimental observations.

An element's standard reduction potential ($E^0$, V) is a regularly applied figure of merit to evaluate its thermodynamic amenability to electron transfer. A case in point is provided by the resistance to reduction of the group 1 monocation, M$^+$ (M = Li, Na, K, Rb, Cs), the endothermicity of which is reflected by their highly negative potentials [i.e. M$_{(aq)}^+$ + e$^-$ → M$_{(s)}$; where M = Li ($E^0$ = −3.04 V vs. SHE), Na (−2.71 V), K (−2.92 V), Rb (−2.98 V), Cs (−2.93 V)][1]. Judged against this criterion alone, the chemical reduction of Li$^+$ to lithium metal is thermodynamically non-viable, even by any of its highly electropositive group 1 congeners. Comparison of $E^0$ values, however, must always take account of the electrochemical experimental conditions[2]. The commonly cited group 1 potentials are determined under standard/aqueous conditions and, thus, heavily biased by the M$^+$ hydration enthalpies ($\Delta H_{Hyd}$), which decline significantly as group 1 is descended (Li$^+$ 506; Na$^+$ 406; K$^+$ 330; Rb$^+$ 310; Cs$^+$ 276 kJ mol$^{-1}$). This observation is nicely borne out by the industrial production of high-purity lithium metal where, and in seeming contradiction of the standard $E^0$ values, the Li$^+$ component is selectively electro-reduced from a molten LiCl−KCl eutectic[3]. In this water-free and primarily ionic system, therefore, the relative decomposition potentials are better estimated from the respective molar Gibbs free energies of formation ($\Delta G^0_m$) of LiCl (−384.4 kJ mol$^{-1}$) and KCl (−408.5 kJ mol$^{-1}$). Although, from this perspective, the electrowinning of Li does follow thermodynamic expectations, similar considerations are not conventionally applied in molecular systems.

The seminal observations of Dye and co-workers' have highlighted the tendency of alkalide ([(L)M]$^+$ M$^-$; where L = typically, a crown ether or cryptand and M = Na, K, Rb, Cs) and electride (i.e. [(L)M]$^+$ e$^-$) species to comproportionate and/or self-reduce[4–7]. In a startling recent advance, Lu and co-workers have shown that the room temperature stable electride, K$^+$[LiN(SiMe$_3$)$_2$]e$^-$ (1)[8], may be directed toward partially selective self-reduction of either the K$^+$ or Li$^+$ component leaving the other in its cationic form[9]. Notably, these processes are triggered by addition of an external Lewis base specific to the group 1 centre that is to maintain its cationic configuration; $viz$, preferential Li$^+$ binding by tris[2-dimethylamino)ethyl]amine (Me$_6$Tren) biases the system toward reduction to potassium, whereas addition of 2,2,2-cryptand induces the deposition of a more lithium-enriched Li/K alloy (Fig. 1).

While Lu's observation appears unique, two systems from molecular low oxidation state magnesium chemistry have recently been reported to yield metallic sodium. Harder's β-diketiminato Mg(0) species, [(BDI*)MgNa]$_2$ (2; BDI* = HC{C($t$-Bu)N(DiPeP)}$_2$; DiPeP = 2,6-(3-pentyl)-phenyl), decomposes to [(BDI*)MgMgMg(BDI*)] to deposit a mirror identified as a 2:1 alloy of metallic Na(0) and Mg(0)[10]. Similarly, we have observed that the bimetallic Mg(I)Na(I) derivative, [{SiN$^{Dipp}$}MgNa]$_2$ (3; [{SiN$^{Dipp}$} = {CH$_2$SiMe$_2$N(Dipp)}$_2$; Dipp = 2,6-$i$-Pr$_2$C$_6$H$_3$)][11], extrudes a pure sodium mirror when treated with a non-reducible base such as THF (Fig. 2a)[12]. Computational studies suggested intramolecular {Mg-Mg}→Na$^+$ electron transfer was initiated in 3 by O→Na coordination and disruption of the otherwise stabilising Na$^+$···arene interactions.

Prompted by the now extensive chemistry arising from Mg−Mg bonded β-diketiminates[13,14], we[15], and others[16], have also sporadically attempted to synthesise analogous Be(I) derivatives. Although a Be−Be

[1]Department of Chemistry, University of Bath, Claverton Down, Bath BA2 7AY, UK. ✉e-mail: cm2025@bath.ac.uk; msh27@bath.ac.uk

bond has very recently been realised in CpBeBeCp[17], our attempt to access a beryllium analogue of compound **3**, by either Li or Na reduction of the 2-coordinate beryllium dianilide, [{SiN$^{Dipp}$}Be] (**4**), resulted in C–H activation of the benzene solvent and isolation of phenylberyllate species, [M({SiN$^{Dipp}$}BePh)] (**5$^M$**, where M = Li or Na; Fig. 2b)[18].

While access to **5$^M$** was proposed to be dependent on the formation of transient Be(I) radical anions, **4** was itself isolated via the initially formed lithium chloroberyllate, [{SiN$^{Dipp}$}BeClLi]$_2$, **6**. The elimination of LiCl required for the isolation of **4** (Fig. 2b) led us to identify compound **6** as a further, but already dimeric, candidate for reduction. In this contribution, therefore, we report the reactivity of **6** with Na, K, Rb and Cs, which, although the isolation of the targeted Be(I) species remains to be achieved, subverts naïve expectation derived from the relevant s-block reduction potentials.

## Results and discussion

### Syntheses

An initial reaction was performed under the conditions applied in the synthesis of **3** between compound **6** and 5 wt.% Na/NaCl in benzene[11]. Filtration, evaporation of the filtrate and crystallisation of the resultant solid from hexane provided colourless crystals of compound **7**. Analysis of a C$_6$D$_6$ solution of **7** by $^1$H and $^{13}$C NMR spectroscopy provided data reminiscent of the starting material (**6**, δ$^9$$_{Be}$ = 8.2 ppm; ω$_{1/2}$ = 314 Hz), while the identification of a broad (ω$_{1/2}$ = 435 Hz) $^9$Be resonance at δ 9.1 ppm confirmed the retention of low coordinate beryllium as a constituent element[19]. Although the absence of any observable $^7$Li NMR signal prompted initial speculation that a Be–Be-bonded species had indeed been isolated, this deduction was overturned by a subsequent single crystal X-ray diffraction experiment (Fig. 3a). This analysis revealed **7** to be a further chloroberyllate

**Fig. 1 | Lu and co-workers' electride system that displays partially selective self-reduction of either Li$^+$ or K$^{+9}$.** Treatment of K$^+$[LiN(SiMe$_3$)$_2$]e$^-$ with Me$_6$Tren leads to preferential encapsulation of Li$^+$ and deposition of a potassium-rich metallic alloy. Reaction of K$^+$[LiN(SiMe$_3$)$_2$]e$^-$ with 2,2,2-crypt results in the formation of a more lithium-enriched Li/K alloy.

derivative, [{SiN$^{Dipp}$}BeClNa]$_2$, derived from **6** by exchange of the lithium counter-cations for sodium. Although the metrics arising from **7** are otherwise unremarkable, its gross structure presents a significant contrast to that of **6**. Whereas the coordination sphere of each lithium in **6** was augmented by η$^6$-engagement with a single N-Dipp group, the larger sodium atoms of **7** enable more symmetrical polyhapto interactions between the alkali metal cations and two pairs of SiN$^{Dipp}$ arene substituents provided by each chloroberyllate moiety (range: Na2-C33 2.783(2)–Na1-C52 3.069(2) Å).

An initial assumption that **7** resulted from simple cation exchange between **6** and the NaCl used to support the alkali metal reducing agent was refuted by the addition of a solution of **6** to a bulk sodium mirror. This reaction also provided compound **7** in high (79%) isolated yield (Fig. 4) indicative of electron transfer from sodium ($E^0$ = −2.71 V) to lithium ($E^0$ = −3.04 V) rather than the anticipated beryllium-centred reduction (Be$^{2+}$$_{(aq)}$ + 2e$^-$ → Be$_{(s)}$ $E^0$ = −1.99 V vs. SHE).

In light of this observation, **6** was reacted under analogous conditions with both a potassium mirror and KC$_8$. Both reactions provided an identical outcome and, after crystallisation from the filtered reaction solutions, the isolation of colourless crystals of compound **8** (Fig. 4). Analysis of **8** by $^1$H, $^{13}$C and $^9$Be (δ$^9$$_{Be}$ = 9.8 ppm; ω$_{1/2}$ = 399 Hz) NMR spectroscopy provided evidence of a structure strongly reminiscent of that of compound **7**. This supposition was verified by the subsequent X-ray diffraction analysis (Fig. 3b), which confirmed the operation of an analogous K$_{(s)}$ + Li$^+$$_{(6)}$ → K$^+$$_{(8)}$ + Li$_{(s)}$ redox process.

Analogous reactions performed between **6** and elemental rubidium and caesium yielded similar outcomes (Fig. 4) and the respective isolation of the Rb and Cs compounds, **9** and **10**, albeit the Cs-based process required prompt work-up to suppress onward reaction and the formation of further, as-yet-unidentified, products. While both compounds again present similar dimeric molecular structures (Fig. 5), the larger Cs cations of **10** facilitate an infinite 2-dimensional network propagated through a sequence of intermolecular Cs···HC close contacts (Supplementary Fig. 33).

The scope of this notably specific reactivity was further explored by sequential treatment of an initial C$_6$D$_6$ solution of **6** with each of the bulk heavier alkali metals. Accordingly, analysis by $^1$H NMR spectroscopy after reaction with a sodium mirror confirmed the generation of compound **7** (Supplementary Figs. 21 and 22). Filtration of this solution and addition to a potassium mirror induced the generation of compound **8**, while the successive extension of this protocol to rubidium and caesium confirmed that the alkali metal redox process could be extended across the entire scope of group 1 elements. As a

**Fig. 2 | Previously reported reduction chemistry of [{SiNDipp}Ae] (Ae = Mg or Be). a** Sodium reduction of [{SiNDipp}Mg] results in the initial formation of the Na(I) Mg(I) species (**3**), which is unstable to sodium metal extrusion when reacted with THF[11,12]; **b** LiCl elimination from the lithium chloroberyllate (**6**) provides [{SiN$^{Dipp}$}Be] (**4**), alkali metal reduction of which results in C–H activation of the benzene solvent to generate the relevant group 1 phenylberyllates (**5$^M$**; M = Li or Na)[18].

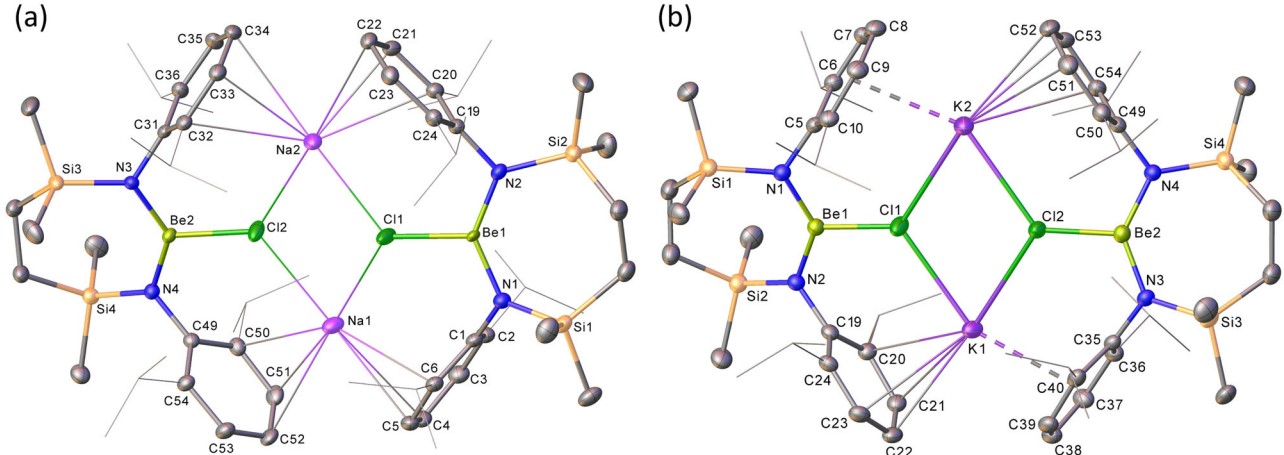

**Fig. 3 | The solid-state structures resulting from the single crystal X-ray diffraction analysis of compounds 7 and 8. a** Displacement ellipsoid plot (30% probability) of compound **7**; **b** displacement ellipsoid plot (30% probability) of

compound **8**. For clarity, hydrogen atoms and solvent (hexane (**7**) and benzene (**8**)) have been omitted and *iso*-propyl substituents are shown as wireframes.

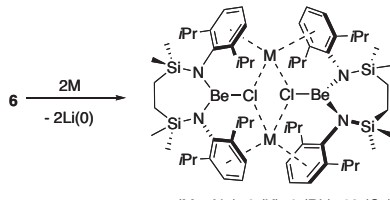

**Fig. 4 | Synthesis of compounds 7–10.** Reactions of the heavier alkali metals (M = Na, K, Rb or Cs) with [{SiN$^{Dipp}$}BeClLi]$_2$ (**6**) result in reduction of the lithium cation and formation of the relevant group 1 chloroberyllates, [{SiN$^{Dipp}$}BeClM]$_2$ (**7** M = Na; **8** M = K; **9** M = Rb; **10** M = Cs). Compounds **7–10** have been characterised by multinuclear NMR spectroscopy and single-crystal X-ray diffraction analysis.

counterpoint to these observations, compound **10** was resistant to the reformation of compound **9** when treated with rubidium (Supplementary Fig. 23). Addition of potassium to **9**, however, yielded a solution of [{SiN$^{Dipp}$}BeClK]$_2$ (**8**) (Supplementary Fig. 25), while no evidence of reaction was observed when this resultant solution was added to either a sodium mirror or an excess of 5 wt% Na/NaCl. This interconversion reactivity is summarised in Fig. 6.

### Theoretical analysis

It is well established that the electrochemical behaviour of transition metal complexes is profoundly affected by changes in coordination number and ligand field strength[20]. In contrast, reduction potential data associated with the less redox-flexible elements of groups 1 and 2 are generally perceived to be invariant. The M′/M$^+$ → M$^{+}$/M interconversion and the maintenance of the Be(II) oxidation level during the synthesis of compounds **7–10**, however, suggests that the susceptibility to reduction of both *s*-block element centres may also be influenced by its local environment. While the resistance to reduction of the diamidoberyllate unit of **6** may be plausibly ascribed to the presence of its chloride substituent, rationalisation of the group 1-redox reactivity requires a more nuanced consideration of both the heterogeneous molecular (solution) M$^+$ and bulk metallic reaction partners.

Non-covalent interactions between alkali metal cations and aromatic π systems are known to play a pivotal role in the regulation of many biological phenomena[21,22]. The importance of polyhapto arene-to-cation binding, for example, during the membrane transport of K$^+$ and the structures and function of many enzymes, has prompted intense interest in the nature and specificity of such interactions[23].

Experiments on gas-phase cluster ions of the form [M(C$_6$H$_6$)$_n$(H$_2$O)$_m$]$^+$, where M = Na or K, for example, have demonstrated that, whereas benzene cannot displace water from the first hydration shell of Na$^+$, the interaction between benzene and K$^+$ induces partial dehydration of the ion[24–26]. Compounds **7–10**, [(SiN$^{Dipp}$)BeClM]$_2$, vary only in the identity of the group 1 atom. Their free energies of formation from compound **6**, therefore, should be quantifiable from the relative strength of the *N*-Dipp···M interactions and the solid-state stabilities of the elemental alkali metals.

Accordingly, density functional theory (DFT) calculations were performed to assess the relative thermodynamic viability of **7–10** via reaction of **6** with the corresponding alkali metal reductant. An all-electron basis set approach was used in the single-point corrections, given that both prior computational studies of C$_6$H$_6$···M interactions by Armentrout and co-workers[27] and our own basis set testing (Supplementary Table 2) revealed that an all-electron basis set yields more accurate M$^+$···arene dissociation energies. Calculations herein are, therefore, reported at the BP86-D3BJ,CPM(Toluene)/ZORA(-SARC-)-def2-TZVPP//BP86/BS1 level of theory, in which the 0th order regular approximation (ZORA) was included in the DFT calculations to account for relativistic effects (see the Supplementary Information for full computational details).

Initial calculations approximated the group 1 reductants as solvent-separated open-shell (doublet) metal atoms. This approach does not account, however, for the heterogeneous nature of the two reactants and the resulting formation energetics (Supplementary Table 3) suggested that lithium reduction (with the exception of the formation of **8**) is endergonic, i.e. thermodynamically unviable and inconsistent with experimental observation. To better replicate the electronic structures and metallic bonding within the bulk metals and following our previous computational studies of the extrusion of sodium upon addition of a weak Lewis base to **3**[12], the group 1 elements were optimised as nonameric $^2$[M$_9$] clusters and used as a fraction in the reaction free energy calculations (Table 1a).

This approach led to a promisingly improved alignment with the experiment, where all cases were identified as exergonic, with the formation of **8** most favoured (ΔG(**8**) = −20.5 kcal mol$^{-1}$). While the formation of the rubidium beryllate **9** (ΔG(**9**) = −17.7 kcal mol$^{-1}$) was also calculated as exergonic relative to **6**, the disparity in free energies between **9** and **8** (ΔΔG(9−8) = +2.8 kcal mol$^{-1}$) suggests an enormous excess of Rb would be necessary to overcome the equilibrium constant of ~0.009. Moreover, although not included in our synthetic study, the small difference in computed reaction energies between **8** and **10**

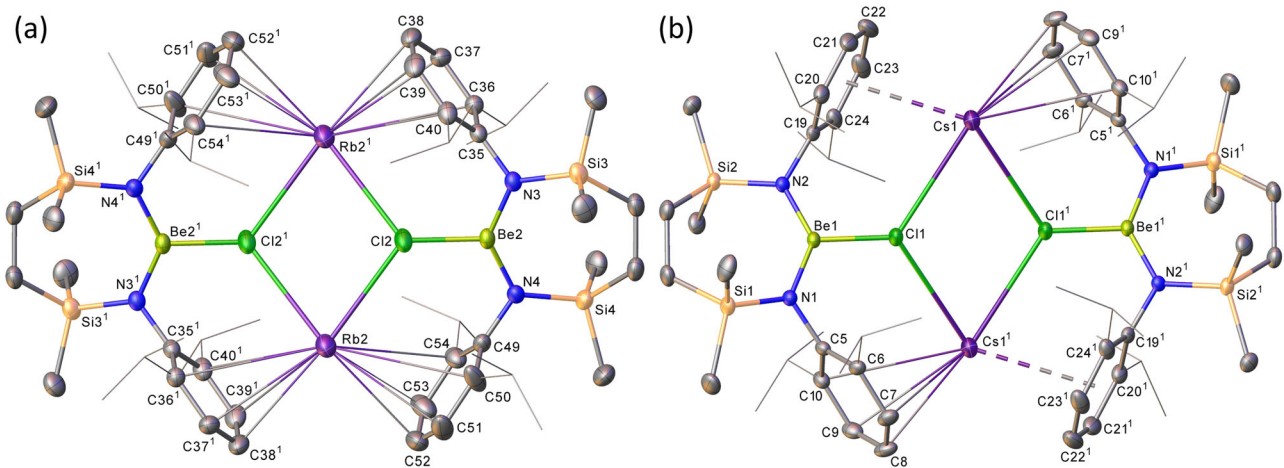

**Fig. 5 | The solid-state structures resulting from the single crystal X-ray diffraction analysis of compounds 9 and 10. a** Displacement ellipsoid plot (30% probability) of the Be2-containing molecule of compound **9**; **b** displacement ellipsoid plot (30% probability) of compound **10**. For clarity, hydrogen atoms and disordered components as well as solvent (benzene in both **9** and **10**) have been omitted. Additionally, for visual ease, *iso*-propyl substituents are shown as wire-frames. Symmetry operations to generate equivalent atoms: (**9**) $^1$ 3/2 − x, 3/2 − y, 1−z; (**10**) $^1$ 1−x, 1−y, 1−z.

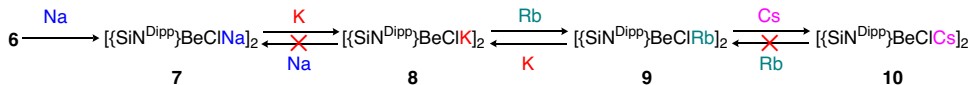

**Fig. 6 | Observed scope of the group 1 redox reactivity.** Only [{SiN$^{Dipp}$}BeClRb]$_2$ (**9**) is amenable to reduction by potassium to its nearest lighter congener. Each species, however, may be sequentially interconverted by treatment of [{SiN$^{Dipp}$}BeClM]$_2$ by the successively heavier group 1 metal.

($\Delta\Delta G(\mathbf{10}-\mathbf{8}) = +0.3$ kcal mol$^{-1}$) implies that the formation of **8** from **10** is thermodynamically feasible.

The third chemical model presented in Table 1b presents reaction energies based on the solvent-separated atomic group 1 metal being empirically corrected with available thermodynamic data for the group 1 elements[28]. Based on a combined experimental and computational Hess cycle, solid-state ($S_{(s)}$) and gas-phase ($S_{(g)}$) entropies were employed along with enthalpies of atomisation, $\Delta_{at}H$ (see Section C3.3 in the Supplementary Information for further details), in an effort to account for the solid-state phase of the alkali metal reactant and by-products. Using this approach, all reduction reactions were calculated to be exergonic, yet the magnitude of exergonicities was of the order **7** ($-17.5$ kcal mol$^{-1}$) ≫ **8** ($-31.7$) > **9** ($-32.2$) > **10** ($-34.5$), which is more consistent with experimental observations. Specifically, the reaction energies of the formation of **8** and **9** are much closer ($\Delta\Delta G(\mathbf{9}-\mathbf{8}) = -0.5$ kcal mol$^{-1}$), correctly identifying that the formation of **9** is viable upon the reaction of **8** with Rb metal, and likewise that **9** is liable to transform to **8** when treated with a K mirror. The improvement in the accuracy of results when incorporating empirical corrections from elemental thermodynamics expresses the importance of varying lattice stabilities within the alkali metals to account for their reactivity. The comparative strengths of the *N*-Dipp⋯M interactions are the subject of ongoing experimental and computational investigations.

In conclusion, we have observed that the order of stability toward the reduction of molecular species comprising arene-to-M$^+$ interactions with other alkali metal reaction partners contradicts the expectation provided by simple consideration of relative reduction potentials. We are continuing to explore the generality of this behaviour as a means to realise otherwise inaccessible main group compounds in which similar M$^+$ encapsulation is intrinsic to the structure and stability of the system. More broadly, we believe these observations may carry significant implications for other areas of scientific and technological importance in which M$^+$ mobility and ease of electrochemical cycling are primary considerations.

## Methods

Synthetic methods, spectroscopic and analytical data for the new compounds, X-ray crystallographic studies and computational details are given in the Supplementary Information.

## Data availability

The X-ray crystallographic coordinates for the structures reported in this study have been deposited at the Cambridge Crystallographic Data Centre (CCDC) under deposition numbers CCDC 2293577 (**7**), 2293578 (**8**), 2293579 (**9**) and 2293580 (**10**). All other data are available from the corresponding authors upon request. Source data for the computational studies are present. Source data are provided in this paper.

## Table 1 | DFT-calculated energetics of formation of 7–10 and extruded Li(0) from 6 and the respective alkali metal reductant M(0)

|                | (a) M$_{9 \text{ (solv.)}}$ | (b) M$_{(s)}$ |            |
|----------------|:---------------------------:|:-------------:|:----------:|
|                | ΔG                          | ΔH            | ΔG         |
| **7** (Na)     | −7.2                        | −17.6         | −17.5      |
| **8** (K)      | −20.5                       | −32.4         | −31.7      |
| **9** (Rb)     | −17.7                       | −31.9         | −32.2      |
| **10** (Cs)    | −20.2                       | −34.4         | −34.5      |

(a) Calculated using a $^2$[M$_9$] cluster. (b) Calculated using an enthalpy/free energy cycle, taking thermodynamic values for the group 1 elements from available experimental data. Calculations were carried out at the BP86-D3BJ,CPM(Toluene)/ZORA(-SARC-)-def2-TZVPP//BP86/BS1 level, with reported energies in kcal mol$^{-1}$.

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

## Acknowledgements

The authors gratefully acknowledge EPSRC (EP/X01181X/1, 'Molecular *s*-block Assemblies for Redox-active Bond Activation and Catalysis: Repurposing the *s*-block as 3*d*-elements') and the University of Bath's Research Computing Group (doi.org/10.15125/b6cd-s854) for their support in this work. This work also used the Isambard 2 UK National Tier-2 HPC Service (http://gw4.ac.uk/isambard/) operated by GW4 and the UK Met Office, and funded by EPSRC (EP/T022078/1).

## Author contributions

K.G.P. carried out the synthesis and characterisation of compounds 7–10 and solved the crystal structures of 8–10. H.Y.L. collected the single crystal X-ray crystallographic data and solved the structure of compound 7. C.L.M. devised, and S.E.N. and H.M.G. carried out and interpreted the theoretical analysis and the DFT calculations. M.F.M. finalised the X-ray structures for publication. M.S.H. coordinated the project and compiled the paper. All the authors discussed the results and contributed to the paper.

## Competing interests

The authors declare no competing interests.
