## [Peer Review File · Nature Communications]

Reviewers' Comments:

Reviewer #1:

Remarks to the Author:

This manuscript by Hill and coworkers describes the unexpected behaviour of beryllium-alkali complexes whereby reduction of a lithium cation can be achieved, contrary to expectations based on potentials vs. SHE, by metallic sodium. Likewise, the sodium cation can be reduced by potassium, potassium cation can be reduced by rubidium, and so on.

This paper is telling us about a very surprising, and fundamental, discovery. Whether it can be extended beyond the present compounds should be ascertained later, but this work certainly defies the established, and apparently erroneous, way of thinking. It constitutes unarguably a great result in main group metal chemistry, and it is worthy of publication.

The manuscript is very well presented, as always with these authors. It reads very easily, and the context, reasoning and main conclusions are superbly presented. It is intelligible to everyone, even to those who are not expert in the field. The results are clear, the trends and conclusions are sound; in particular, the reactivity summarised in Scheme 4 is nothing but exquisite.

Queries I have relate to the use of beryllium. It is a highly toxic element, and to become useful, the chemistry presented herein would need to be rid of it. Is it conceivable to extend it to the next element in group 2, magnesium? Have attempts been made? The other thing, and this is mentioned in the final part of the MS, how essential are secondary interactions in this reactivity? In the case of cesium, how relevant are the Cs...H-C close contacts that are mentioned?

My only criticism is the scale of these reactions. Tiny amounts of the complexes were isolated, as low as 11-16 mg. Compound 10 is not stable under vacuum, and no yield is reported for it. I would really have liked to see these reactions scaled up, in order to see if several hundred milligrams, ie a synthetically useful amount, can be produced.

Other than this, there is little to comment. I would however recommend to seek the advice of an expert in theoretical computations to assess the work presented herein. From my non-expert eyes, though, it looks perfectly fine.

In summary, the work is great, and the manuscript can be accepted for publication after the items above, in particular in link with the scales of the reactions, have been addressed.

Reviewer #2:

Remarks to the Author:

Professor Michael Hill and coworkers have presented a series of reduction reactions that behave contrary to the typically understood standard reduction potentials. The reduction of a dimeric beryllium(II) species co-crystallized with LiCl has yielded the reduction to elemental Li, which is conventionally understood to be impossible by chemical means. A series of 'impossible' reduction reactions with Group(I) elements have yielded the corresponding Group(I) halide complexes of the ancillary beryllium species. This non-intuitive reactivity is supported by a series of quantum chemical calculations which shed light on the thermodynamic feasibility of the findings presented. The chemistry described is well-characterized, and the tone of the article is of a high degree of scholarship.

Despite the new molecules presented in the paper not being of particular interest, their method of access is extremely interesting, and I believe that this article will help guide future studies in the synthesis of low-valent s- and p-block molecules. I would recommend the article for publication in Nature communications.

The supporting spectroscopic data is in full agreement with the conclusions reached. The X-ray crystallographic data is well treated. I am unable to provide an assessment of the computational methods as this is out of my field of expertise. Elemental Analysis for compounds 8 and 10 should be provided if possible.

Reviewer #3:

Remarks to the Author:

This contribution by Hill et al. relates to the reaction of lithium chloroberyllate, $[\text{LBeClLi}]_2$ [$\text{L} = \{\text{CH}_2\text{SiMe}_2\text{N}(\text{dipp})\}_2$; $\text{dipp} = 2,6\text{-iPr}_2\text{C}_6\text{H}_3$], containing a bulky chelate diamido ligand L with the heavier alkali metals M (M = Na, K, Rb, Cs). Unexpectedly and in contrast to the difference in standard redox potentials, lithium is reduced to give alkali metal chloroberyllate $[\text{LBeClM}]_2$. Heavier alkali metals also reduced its lighter congener. All new compounds have been isolated and fully characterized including single crystal X-ray diffraction. The structure of the products exhibit typical interactions of M ions with both chlorines's p-electrons and pi-interactions to the aromatic rings, probably the most important structure-determining element in molecular alkali metal compounds. DFT calculations have been performed to explain the observed reaction pattern, while thermodynamic considerations allowed to identify enthalpies of atomization of bulk alkali metal as essential to explain the experimental observations.

The work presented is a fine example of a highly competently performed study in the area of s-block metal chemistry, now considered to be one of the rapidly developing areas of topical main group chemistry and can be recommended for publication in Nature Commun..

Reviewer #4:

Remarks to the Author:

The manuscript entitled "Alkali Metal Reduction of Alkali Metal Cations" by McMullin, Hill and co-workers presents the synthesis and unanticipated reactivity of a series of ligated alkali metal beryllium chloride compounds. The lighter alkali metal cations in this series of compounds are found to be reduced by heavier alkali metals in a manner that appears to contradict the expectations one would have on the basis of the standard reduction potentials that are used by chemists - and chemistry students - everywhere to rationalize chemical reactivity. This remarkable behavior is sufficiently counter-intuitive and noteworthy that it will be of interest to the wide readership of Nature Communications. Although there are some other somewhat related "one-off" reactions (noted by the authors) this investigation is the first series that permits systematic investigation of the kind presented. The authors prepare and characterize each of the compounds using a battery of analytical methods that are appropriate and convincing. The methodology of the experiments is certainly of a standard and described in a level of detail that would allow other to reproduce the compounds. Similarly, the single-crystal XRD investigations appear to have been conducted in an appropriate manner and the high-quality data are similarly conclusive. The authors explore a series of computational methods to identify reasonable approaches to use to rationalize the observed chemical reactivity and the methods they identify are consistent with the observed results and attempt to address the heterogeneity in the actual systems. The conclusions drawn by the authors follow clearly from the experimental and computational results. This reviewer is comfortable with the manuscript being published essentially as it is.

Although it is certainly not necessary for this publication, this reviewer wonders if the authors have insights about the mechanism of the reaction?

Given that the dimeric core structure of 6 is clearly maintained in each of the products, have the

authors attempted to isolate any of the mixed-alkali-metal "intermediate" compounds using a sub-stoichiometric amount of heavier alkali-metal?

Thank you very much for your email of 20/11/23 to inform us that our manuscript (NCOMMS-23-44176-T)

Title: Alkali Metal Reduction of Alkali Metal Cations

Authors: Kyle G. Pearce, Han-Ying Liu, Samuel E. Neale, Hattie M. Goff, Mary F. Mahon, Claire L. McMullin and Michael S. Hill

had been positively assessed by four expert referees as suitable for publication in *Nature Communications*. We are particularly pleased that each referee clearly recognised the surprising and very fundamental nature of the chemistry reported. Although each reviewer made some very complimentary and gratifying comments on the quality of our work and the clarity of its presentation, they also raised a number of pertinent issues for clarification and consideration prior to finalisation of our manuscript. We are, thus, more than happy to respond to each of the reviewers, in turn, below.

We have also made the requested editorial and formatting adjustments, both to our revised manuscript and the associated Supplementary Information. Changes of this nature are comprehensively summarised in the completed Author Checklist, which was provided as an attachment to your previous email.

Reviewer 1

This manuscript by Hill and coworkers describes the unexpected behaviour of beryllium-alkali complexes whereby reduction of a lithium cation can be achieved, contrary to expectations based on potentials vs. SHE, by metallic sodium. Likewise, the sodium cation can be reduced by potassium, potassium cation can be reduced by rubidium, and so on.

This paper is telling us about a very surprising, and fundamental, discovery. Whether it can be extended beyond the present compounds should be ascertained later, but this work certainly defies the established, and apparently erroneous, way of thinking. It constitutes unarguably a great result in main group metal chemistry, and it is worthy of publication.

The manuscript is very well presented, as always with these authors. It reads very easily, and the context, reasoning and main conclusions are superbly presented. It is intelligible to everyone, even to those who are not expert in the field. The results are clear, the trends and conclusions are sound; in particular, the reactivity summarised in Scheme 4 is nothing but exquisite.

We are, of course, very pleased by the tenor of the referee's overall opinion.

Queries I have relate to the use of beryllium. It is a highly toxic element, and to become useful, the chemistry presented herein would need to be rid of it. Is it conceivable to extend it to the next element in group 2, magnesium? Have attempts been made?

We have earlier reported comparable and related magnesium chemistry, from which the current work grew, in the paper cited as reference 11 (*J. Am. Chem. Soc.* 2021, **143**(42): 17851-17856). In light of

the current work, we are re-evaluating this chemistry through variation of the spectator ligand and with alternative alkaline earth elements. Similar research to generalise the group 1 redox chemistry to a wider range of molecular p-block systems and to understand the potential scope of this reactivity is also ongoing, albeit this is as yet unready for dissemination.

The other thing, and this is mentioned in the final part of the MS, how essential are secondary interactions in this reactivity? In the case of cesium, how relevant are the Cs...H-C close contacts that are mentioned?

We do not consider this solid-state structural feature is likely to be relevant to the solution-based reduction chemistry central to this research. This comment is included simply to highlight the structural contrast induced in the solid-state structure by inclusion of the larger Cs⁺ cation.

My only criticism is the scale of these reactions. Tiny amounts of the complexes were isolated, as low as 11-16 mg. Compound 10 is not stable under vacuum, and no yield is reported for it. I would really have liked to see these reactions scaled up, in order to see if several hundred milligrams, ie a synthetically useful amount, can be produced.

The referee makes a pertinent observation about the scale of the reactions reported. Due to the notable toxicity of beryllium and its compounds (not to mention the hazards associated with handling of larger quantities of rubidium and caesium), we prefer to deliberately work on the micro scale so as to limit, as far as possible, the risk of ambient proliferation. As all four reviewers appreciate, we view the beryllium species reported in the current work as specific expedients with which to demonstrate some previously unappreciated fundamental chemistry of the alkali metals. We do, however, submit samples of all new crystalline compounds to an external supplier for elemental analysis. On this occasion, no corroborative data could be obtained, presumably because of the instability of the compound and resultant decomposition in transit. A similar justification is provided for compound **8**, although our 'closest result' from multiple attempts for this compound is now included in the revised ESI for this compound.

Other than this, there is little to comment. I would however recommend to seek the advice of an expert in theoretical computations to assess the work presented herein. From my non-expert eyes, though, it looks perfectly fine.

In summary, the work is great, and the manuscript can be accepted for publication after the items above, in particular in link with the scales of the reactions, have been addressed.

We, again, very much appreciate the referee's opinion of our work.

Reviewer 2

Professor Michael Hill and coworkers have presented a series of reduction reactions that behave contrary to the typically understood standard reduction potentials. The reduction of a dimeric beryllium(II) species co-crystallized with LiCl has yielded the reduction to elemental Li, which is conventionally understood to be impossible by chemical means. A series of 'impossible' reduction reactions with Group(I) elements have yielded the corresponding Group(I) halide complexes of the ancillary beryllium species. This non-intuitive reactivity is supported by a series of quantum chemical calculations which shed light on the thermodynamic feasibility of the findings presented. The chemistry described is well-characterized, and the tone of the article is of a high degree of scholarship.

Despite the new molecules presented in the paper not being of particular interest, their method of access is extremely interesting, and I believe that this article will help guide future studies in the synthesis of low-valent s- and p-block molecules. I would recommend the article for publication in Nature communications.

We are again very grateful for the positive consideration of this referee.

The supporting spectroscopic data is in full agreement with the conclusions reached. The X-ray crystallographic data is well treated. I am unable to provide an assessment of the computational methods as this is out of my field of expertise. Elemental Analysis for compounds **8** and **10** should be provided if possible.

Please see our response to Reviewer 1 with regard to the absence of corroborative elemental analysis for compounds **8** and **10**.

Reviewer 3

This contribution by Hill et al. relates to the reaction of lithium chloroberyllate, $[\text{LBeClLi}]_2$ [$\text{L} = \{\text{CH}_2\text{SiMe}_2\text{N}(\text{dipp})\}_2$; $\text{dipp} = 2,6\text{-iPr}_2\text{C}_6\text{H}_3$], containing a bulky chelate diamido ligand L with the heavier alkali metals M (M = Na, K, Rb, Cs). Unexpectedly and in contrast to the difference in standard redox potentials, lithium is reduced to give alkali metal chloroberyllate $[\text{LBeClM}]_2$. Heavier alkali metals also reduced its lighter congener. All new compounds have been isolated and fully characterized including single crystal X-ray diffraction. The structure of the products exhibit typical interactions of M ions with both chlorines's p-electrons and pi-interactions to the aromatic rings, probably the most important structure-determining element in molecular alkali metal compounds. DFT calculations have been performed to explain the observed reaction pattern, while thermodynamic considerations allowed to identify enthalpies of atomization of bulk alkali metal as essential to explain the experimental observations.

The work presented is a fine example of a highly competently performed study in the area of s-block metal chemistry, now considered to be one of the rapidly developing areas of topical main group chemistry and can be recommended for publication in Nature Commun.

We again thank the referee for their very complimentary assessment of our contribution.

Reviewer 4

The manuscript entitled "Alkali Metal Reduction of Alkali Metal Cations" by McMullin, Hill and co-workers presents the synthesis and unanticipated reactivity of a series of ligated alkali metal beryllium chloride compounds. The lighter alkali metal cations in this series of compounds are found to be reduced by heavier alkali metals in a manner that appears to contradict the expectations one would have on the basis of the standard reduction potentials that are used by chemists - and chemistry students - everywhere to rationalize chemical reactivity. This remarkable behavior is sufficiently counter-intuitive and noteworthy that it will be of interest to the wide readership of Nature Communications. Although there are some other somewhat related "one-off" reactions (noted by the authors) this investigation is the first series that permits systematic investigation of the kind presented. The authors prepare and characterize each of the compounds using a battery of analytical methods that are appropriate and convincing. The methodology of the experiments is certainly of a standard and described in a level of detail that would allow other to reproduce the compounds. Similarly, the single-crystal XRD investigations appear to have been conducted in an appropriate manner and the high-quality data are similarly conclusive. The authors explore a series of computational methods to identify reasonable approaches to use to rationalize the observed chemical reactivity and the methods they identify are consistent with the observed results and attempt to address the heterogeneity in the actual systems. The conclusions drawn by the authors follow clearly from the experimental and computational results. This reviewer is comfortable with the manuscript being published essentially as it is.

We thank the referee for their very positive comments and appreciation of our work.

Although it is certainly not necessary for this publication, this reviewer wonders if the authors have insights about the mechanism of the reaction?

As outlined above, we are at present attempting to extend this chemistry in a number of directions that ultimately may provide more concrete mechanistic insight. As the referee recognises, our initial attempt

to provide a coherent thermochemical assessment of this reactivity has required some lateral thinking, and the adoption of a hybrid approach, which utilises computational analysis in conjunction with existing experimental physical data. This approach is, to the best of our knowledge, also completely novel and something that we will continue to elaborate.

Given that the dimeric core structure of **6** is clearly maintained in each of the products, have the authors attempted to isolate any of the mixed-alkali-metal "intermediate" compounds using a sub-stoichiometric amount of heavier alkali-metal?

This is a very appealing thought, although control of reaction stoichiometry will be very difficult to achieve. We do not (yet) have any experimental support for this hypothesis, but certainly do not discount the possibility.

With our submission we provide the complete revised manuscript with changes indicated by yellow highlighting. Experimental details, NMR spectra, a summary of the X-ray analysis and details of the computational analysis are provided in pdf format in the revised Supplementary Information. The coordinates provided by the computational studies are provided as a separate readable .xyz file labelled 'Source data_computed_structures' and Figures 1 - 6 are provided either as Chem Draw files (1, 2, 4 and 6) or high resolution .ppt (3 and 5) files. We also include the fully annotated Author Checklist and the completed editorial policy checklist. The checkCIF and CIF files for the X-ray studies of compounds **7 - 10** were included with our original submission and the CIFs have been deposited with the Cambridge Structural Database as CCDC 2182615 - 2182617.

We very much hope that you agree that our work is now suited to publication in *Nature Communications*.